# Transcriptional and Epigenetic Alterations in the Progression of Non-Alcoholic Fatty Liver Disease and Biomarkers Helping to Diagnose Non-Alcoholic Steatohepatitis

**DOI:** 10.3390/biomedicines11030970

**Published:** 2023-03-21

**Authors:** Yalan Zhu, He Zhang, Pengjun Jiang, Chengxia Xie, Yao Luo, Jie Chen

**Affiliations:** Department of Laboratory Medicine, West China Hospital, Sichuan University, Chengdu 610041, China

**Keywords:** non-alcoholic fatty liver disease, non-alcoholic steatohepatitis, DNA methylation, logistic regression, machine learning

## Abstract

Non-alcoholic fatty liver disease (NAFLD) encompasses a broad spectrum of conditions from simple steatosis (non-alcoholic fatty liver (NAFL)) to non-alcoholic steatohepatitis (NASH), and its global prevalence continues to rise. NASH, the progressive form of NAFLD, has higher risks of liver and non-liver related adverse outcomes compared with those patients with NAFL alone. Therefore, the present study aimed to explore the mechanisms in the progression of NAFLD and to develop a model to diagnose NASH based on the transcriptome and epigenome. Differentially expressed genes (DEGs) and differentially methylated genes (DMGs) among the three groups (normal, NAFL, and NASH) were identified, and the functional analysis revealed that the development of NAFLD was primarily related to the oxidoreductase-related activity, PPAR signaling pathway, tight junction, and pathogenic *Escherichia coli* infection. The logistic regression (LR) model, consisting of *ApoF*, *THOP1*, and *BICC1*, outperformed the other five models. With the highest AUC (0.8819, 95%CI: 0.8128–0.9511) and a sensitivity of 97.87%, as well as a specificity of 64.71%, the LR model was determined as the diagnostic model, which can differentiate NASH from NAFL. In conclusion, several potential mechanisms were screened out based on the transcriptome and epigenome, and a diagnostic model was built to help patient stratification for NAFLD populations.

## 1. Introduction

Non-alcoholic fatty liver disease (NAFLD) is now recognized as the most common cause of chronic liver disease, with a global prevalence of 25% [1]. It encompasses a broad spectrum of conditions, from simple steatosis (non-alcoholic fatty liver (NAFL)) to non-alcoholic steatohepatitis (NASH), which is characterized by necroinflammation and faster fibrosis progression than NAFL. Because of its high prevalence, NAFLD is now the most rapidly growing cause of liver-related mortality and morbidity worldwide [2]. Furthermore, there is currently no approved specific therapy for NAFLD, and lifestyle changes such as exercise and dietary modifications remain the mainstream treatment for NAFLD [3]. Compared with NAFL, patients with NASH have an increased risk of adverse hepatic outcomes such as cirrhosis, liver failure, and hepatocellular carcinoma, and also carry a higher risk of non-liver adverse outcomes including cardiovascular diseases and type 2 diabetes mellitus (T2DM) [4,5]. Thus, the differentiation of NASH from NAFL is a key issue for patients with NAFLD [6]. To date, liver biopsy remains the gold standard to differentiate NASH from NAFL. The histopathologic features of NASH generally include the presence of liver steatosis, inflammation, hepatocellular injury, and different degrees of fibrosis [7]. However, these features of NASH are not manifested in equivalent proportions in every biopsy, and no single feature by itself is diagnostic, making the diagnosis difficult at times [8]. Furthermore, its well-known limitations such as the dependence on pathologist experiences and the discrepancy between pathologists have not been solved either. Other methods, either relying on a “physical” approach based on the measurement of liver stiffness or a “biological” approach based on the quantification of biomarkers in the serum samples have been developed for NASH diagnosis, but none have been widely accepted yet in routine practice due to their limited sensitivity or specificity [6]. Therefore, novel objective molecular biomarkers are urgently needed to assist in standardizing and improving the diagnosis of NASH.

Epigenetics is characterized by heritable and reversible changes in gene expression, and it affects the phenotype by regulating gene transcription, without changing the primary DNA sequence [9]. DNA methylation, as the most common epigenetic modification, is closely related to the transcriptional regulation of genes and maintains the stability of the genome [10]. The function of DNA methylation seems to vary with different genomic contexts, but the gene expression level is usually inversely associated with the DNA methylation level [11]. The inverse association between methylation and transcription has been previously reported in NAFLD studies. Tissue repair genes, such as fibroblast growth factor receptor 2 (*FGFR2*) and caspase 1 (*CASP1*), were hypomethylated and high-expressed, whereas a gene in one-carbon metabolism, methionine adenosyl methyltransferase 1A (*MAT1A*), which generates SAM, was hypermethylated and low-expressed in liver biopsies from patients with advanced NAFLD [12]. In recent years, an ever-increasing number of high-throughput omics technologies including transcriptomics and epigenomics have been developed to explore the pathogenesis of NAFLD, as well as the establishment of diagnostic biomarkers [13]. Differential gene expression and significant alterations in DNA methylation are observed in the progression of NAFLD, including genes involved in glucose, lipid, or acetyl-coenzyme A (CoA) metabolism; insulin-like signaling; cellular division; immune function; mitochondrial function; and so on [13]. Several potential diagnostic biomarkers such as the circulating micro-RNAs and the altered DNA methylation sites in peripheral blood leukocytes have also been reported [14,15]. However, most previous studies have merely focused on either gene expression or methylation data, and few diagnostic biomarkers were based on multi-omics.

In the present study, we systematically analyzed the gene expression and DNA methylation data in the occurrence and development of NAFLD. The molecular mechanisms and functional pathways in the progression of NAFLD have been explored in transcriptional and epigenetic aspects, respectively. Using different methods, six diagnostic models based on both abnormally methylated and differentially expressed genes between NAFL and NASH were constructed and compared. The workflow diagram is shown in Figure 1. Our study would further assist in understanding the pathogenesis and patient stratification of NAFLD.

## 2. Materials and Methods

### 2.1. Data Source and Data Processing

The gene expression data and the corresponding DNA methylation data of NAFLD patients were obtained from the Gene Expression Omnibus (GEO, https://www.ncbi.nlm.nih.gov/geo/, accessed on 15 March 2022), and four datasets were included (Table 1). For the RNA transcriptome data, a total of 126 patients (28 normal, 49 NAFL, and 49 NASH) were retrieved from GSE48452 and GSE31803. DNA methylation data (Infinium Human Methylation 450 k) were downloaded from GSE48325 and GSE49542, including 118 patients (34 normal, 45 NAFL, and 39 NASH). Strict criteria were adopted when selecting the samples: patients without a clear histological diagnosis were excluded, and patients who received additional clinical treatments such as bariatric surgery were also excluded.

Considering the impact of different data processing methods, the raw cell files of the gene expression data were downloaded and processed with the R package oligo v.1.58.0 in the same way. Meanwhile, the R package ChAMP v.2.24.0 was used for the methylation analysis. The β value, ranging from 0 (unmethylated) to 1 (fully-methylated), was selected to represent the methylation level of each probe. In addition, the R package impute v.1.68.0, with its K-nearest neighbor (KNN) imputation procedure, was applied to impute the missing values in all datasets. The batch effects between different datasets were also adjusted by the ComBat function in the R package sva v.3.42.0, which allows users to adjust for batch effects in datasets using an empirical Bayes framework.

### 2.2. Identification of DEGs and DMGs in the Progression of NAFLD

To search for critical genes for NAFLD development, we identified differentially expressed genes (DEGs) and differentially methylated genes (DMGs) among the three groups (normal, NAFL, and NASH), seperately. The R package limma v. 3.50.0 was applied to screen DEGs, while DMGs were detected by the ChAMP.DMP function. To improve the accuracy, significant cut-off values of false discovery rate (FDR) < 0.01 and |log2 fold change (FC)| > 0.5 were used to identify DEGs, and FDR < 0.01 and |log2 fold change (FC)| > 0.1 were utilized to determine DMGs.

### 2.3. Functional Enrichment Analysis

For the sake of exploring the possible mechanisms involved in the progression of NAFLD, we conducted the following analyses with the R package clusterProfiler v.4.2.2. Gene ontology analysis (GO) was used for annotating DEGs and DMGs, and Kyoto Encyclopedia of Genes and Genomes (KEGG) was used to perform the pathway enrichment analysis. FDR < 0.05 was set as the threshold value. Moreover, the search tool for the Retrieval of Interacting Genes (STRING) database (version 11.5) was used to evaluate the protein-protein interaction (PPI) information. The interaction score was set at 0.7. MCODE was conducted to screen modules of the PPI network with the degree cutoff, k-core, node score cutoff, and max depth set at 2, 2, 0.2, and 100, respectively, in Cytoscape software (version 3.9.1).

### 2.4. Construction and Validation of the Diagnostic Model

The timely identification of high-risk individuals plays a vital role in clinical practice, so we shifted focus to the construction of the NASH diagnostic model. To provide a robust model, we combined gene expression and methylation data. Genes that were reversely correlated (hypomethylated-high expressed or hypermethylated-low expressed) were identified based on DEGs and DMGs, and were regarded as the input variables in the succeeding study. To avoid overfitting and to simplify the model, least absolute shrinkage and selection operator (LASSO) regression was initially utilized to filter the variables. The key parameter λ was determined by ten-fold cross validation and λ_1se was selected in this study. In addition to the LASSO regression, another five popular methods, including logistic regression (LR), random forest (RF), support vector machine (SVM), extreme gradient boosting (XGBoost), and k-nearest neighbor (KNN), were further conducted to construct the diagnostic model. All of the models were realized by the corresponding R packages: glmnet v.4.1-3, randomForest v.4.7-1.1, e1071 v.1.7-9, xgboost v.1.6.0.1, and kknn v.1.3.1. The hyperparameters of these methods were determined by grid search or cross validation based on the R package caret v.6.0-90. For each model, the Youden index was calculated to determine the optimal cut-off values. Area under curve (AUC), sensitivity, specificity, positive predictive value (PPV), and negative predictive value (NPV) were adopted to evaluate the diagnostic ability of the model. The external dataset, GSE167523 (51 NAFL and 47 NASH), was used as the testing cohort to further assess the diagnostic ability of the above models. 

### 2.5. Development and Assessment of the Nomogram

The R package rms v.6.2-0 was applied to build a nomogram to visualize the final model. Furthermore, calibration curve and decision curve analysis (DCA) were employed to weigh the calibration and clinical applicability of the nomogram.

### 2.6. Validation of the Expression Pattern of the Model Genes in the Testing Cohort

The expression pattern of the model genes was further verified in the testing dataset. The Wilcoxon rank-sum test was used to identify the differential expression levels of the model genes between the NASH and NAFL groups. A two-sided *p* < 0.05 was considered statistically significant. 

### 2.7. Development of NAFL and NASH in Mice

Wild-type male C57BL/6J mice (8 weeks of age) were acquired from Beijing HFK Bio-Technology. The mice were housed in a specific pathogen free environment with a stationary temperature at 22 ± 1 °C on a 12 h dark/light cycle. After 1 week of adaptive feeding, the mice were randomly divided into two groups: high-fat diet (HFD, 60% calories from fat purchased from Research Diet, USA) only and HFD with CCl4 (Sigma-Aldrich, Saint Louis, MO, USA, 289116). CCl4 at a dose of 0.2 μL (0.32 μg)/g of body weight was injected intra-peritoneally once per week, starting simultaneously with the diet administration. After 10 weeks of treatment, liver tissues were collected and processed for histological analysis. NAFLD activity score (NAS) and disease stage were evaluated by an expert pathologist according to the NASH CRN scoring system (Appendix A). Pictures were taken from representative areas showing steatosis, lobular inflammation, and hepatocyte ballooning, in consultation with the pathologist (Appendix A). All of the animal experiments were approved by the Animal Care and Use Committee of West China Hospital, Sichuan University.

### 2.8. Exploration of the Model Genes Expression Levels in Mice Using qRT-PCR

The total RNA was extracted from the liver tissues of the mice using the Trizol reagent (Thermofisher, Singapore, Cat No. 15596026) following the manufacturer’s instructions. The cDNA reverse transcription kit (Accurate Biology, Cat No. AG11711) was used to reverse transcribe RNA, and the SYBR Green Premix Pro Taq HS qPCR kit (Accurate Biology, Cat No. AG11701) was utilized to amplify the resulting cDNA. The samples were detected using an ABI 7500 Real-Time PCR System. The 2^(−∆∆Ct)^ method was adopted to calculate the expression of the genes relative to the housekeeping gene β-Actin. The primers used for qRT-PCR are shown in Appendix A.

## 3. Results

### 3.1. Identification of DEGs and DMGs in the Progression of NAFLD

For the gene expression data, the DEGs were regularly observed during the whole progression of NAFLD: 45 upregulated genes and 52 downregulated genes were identified in the normal vs. NAFL group, respectively; 135 upregulated genes and 23 downregulated genes were identified in the NAFL vs. NASH group, respectively; and 78 upregulated genes and 31 downregulated genes were identified in the normal vs. NASH group, respectively (Appendix A). In contrast, the DMGs were mainly observed in the latter period of NAFLD: no significant difference was observed in the normal vs. NAFL group; 650 hypermethylated genes and 377 hypomethylated genes were identified in the NAFL vs. NASH group, respectively; 195 hypermethylated genes and 99 hypomethylated genes were identified in the normal vs. NASH group, respectively (Appendix A). The volcano plots show the distribution of DEGs and DMGs (Figure 2a–f).

### 3.2. Functional Enrichment Analysis of DEGs

Regarding DEGs, the GO analysis showed that a total of 3, 13, and 24 GO terms were obtained in the normal vs. NAFL group, the NAFL vs. NASH group, and the normal vs. NASH group, respectively (Appendix A). Of note, one overlapping GO term (oxidoreductase activity, acting on paired donors, with the incorporation or reduction of molecular oxygen) was identified between the normal vs. NAFL group and the normal vs. NASH group, and nine overlapping GO terms (extracellular matrix structural constituent, extracellular matrix structural constituent conferring tensile strength, platelet-derived growth factor binding, glycosaminoglycan binding, etc.) were observed between the NAFL vs. NASH group and the normal vs. NASH group (Figure 2g). In addition, the KEGG pathway analysis found 2, 6, and 3 pathways significantly enriched in the normal vs. NAFL group, the NAFL vs. NASH group, and the normal vs. NASH group (Appendix A). Similarly, one overlapping pathway (PPAR signaling pathway) enriched in both the normal vs. NAFL group and the normal vs. NASH group, and one overlapping pathway (ECM-receptor interaction) was identified between the NAFL vs. NASH group and the normal vs. NASH group (Figure 2h). The PPI network map of the DEGs had 107 nodes and 64 edges, and then the network was imported into the Cytoscape software to perform module analysis, in which the DEGs were constructed into four modules (Appendix A).

### 3.3. Functional Enrichment Analysis of DMGs

Meanwhile, a total of 32 GO terms (22 GO terms in the NAFL vs. NASH group and 10 GO terms in the normal vs. NASH group) and 15 KEGG pathways (10 pathways in the NAFL vs. NASH group and 5 pathways in the normal vs. NASH group) were obtained for the DMGs (Appendix A). Both the GO and KEGG results were highly overlapped between the two groups. The GO terms are mainly related to “GTPase regulator activity”, “GTPase activator activity”, “nucleoside-triphosphatase regulator activity”, and “guanyl-nucleotide exchange factor activity”, and the KEGG results indicate that DMGs were mainly involved in “tight junction”, “regulation of actin cytoskeleton”, “pathogenic *Escherichia coli* infection”, and “chemical carcinogenesis-DNA adducts” (Figure 2i,j). The PPI network map of DMGs had 288 nodes and 124 edges, and the DMGs were constructed into six modules (Appendix A).

### 3.4. Construction and Validation of the Diagnostic Model

The diagnostic model was generated based on combined gene expression and methylation data. LASSO regression was initially applied to filter the variables, and another five methods were used to build the diagnostic model. The detailed hyperparameters are summarized in Appendix A.

Based on DEGs and DMGs, a total of 21 genes (16 hypomethylated-high expressed genes and 5 hypermethylated-low expressed genes) were identified, and four genes (*THOP1*, *ApoF*, *BICC1*, and *CCDC146*) were selected as the final input variables by LASSO regression (Appendix A). In the training cohort, all six methods exhibited an extraordinary diagnostic performance (Table 2). The highest AUC was found in the RF model and XGBoost model (AUC = 1.0000, 95%CI: 1.0000–1.0000), followed by the KNN model (AUC = 0.9842, 95%CI: 0.9677–1.0000), the LR model (AUC = 0.9792, 95%CI: 0.9575–1.0000), the SVM model (AUC = 0.9775, 95%CI: 0.9556–0.9994), and the LASSO model (AUC = 0.9733, 95%CI: 0.9476–0.9991). A reduction in performance was inevitable and was considered acceptable in the testing cohort, which contained heterogeneous patient data. Despite all of this, the LR model still reached an AUC of 0.8819 (95%CI: 0.8128–0.9511), higher than the five other models (SVM: AUC = 0.8623, 95%CI: 0.7868–0.9378; KNN: AUC = 0.8502, 95%CI: 0.7707–0.9298; RF: AUC = 0.8454, 95%CI: 0.7695–0.9214; XGBoost: AUC = 0.8256, 95%CI: 0.7455–0.9058; LASSO: AUC = 0.8052, 95%CI: 0.7192–0.8911) (Figure 3f and Appendix A). Therefore, the LR model was eventually determined as the optimal model for diagnosing NASH and was taken into further study. 

### 3.5. Development and Assessment of the Nomogram

Three genes (*ApoF*, *THOP1*, and *BICC1*) were screened out as the model genes by the LR model (*p* < 0.05). Diagnostic scores were calculated using the following formula.
logit (P = NASH) = 0.5470 − (1.5909 × *THOP1* expression level) − (1.3167 × *ApoF* expression level) + (3.9034 × *BICC1* expression level)

Based on the LR model, a nomogram was built to predict the risk score of individual patients, and the three model genes were used as parameters in the nomogram (Figure 3c). The predicted NASH probability was compared to the actual NASH probability in the calibration curve, and a high level of consistency was observed (Figure 3d). Moreover, the DCA curve showed a net benefit across the whole range of threshold probabilities, indicating that the nomogram was feasible to make beneficial clinical decisions (Figure 3e).

### 3.6. Validation of the Expression Pattern of the Model Genes in the Testing Cohort

In the training cohort of NASH patients, significantly high DNA methylation and low expression levels were noted for *ApoF* and *THOP1*, while a low DNA methylation and high expression level was observed for *BICC1* (*p* < 0.001) (Figure 4e–j). To further validate the expression pattern of the three model genes, these genes were selected from the GSE167523 testing cohort. As shown in Figure 4k–m, *ApoF* and *THOP1* exhibited a significantly lower expression in the NASH group than in the NAFL group, whereas *BICC1* exhibited a higher expression in the NASH group than in the NAFL group (*p* < 0.001). Altogether, the consistent results between different cohorts demonstrated that the expression levels of the three genes were reliable and useful for constructing the diagnostic model.

### 3.7. Exploration of the Expression Pattern of the Model Genes in the Mouse Model 

The model genes selected by the LR model might help in improving the understanding of the disease pathogenesis. Thus, we preliminarily explored the expression pattern of the three genes in the mouse model. Our results indicate that the NAFL group exhibited a higher *THOP1* expression level than the NASH group (*p* < 0.05), which was consistent with the results we obtained from the human cohorts (Figure 5). However, no meaningful findings were observed for the other two genes. The original data are provided in Appendix A. 

## 4. Discussion

NAFLD is the hepatic manifestation of the metabolic syndrome, representing a substantial health and economic burden worldwide. NASH, the progressive form of NAFLD, may culminate into cirrhosis and hepatocellular carcinoma, and is presently a leading cause of liver transplant [16]. Because of the poor prognosis of NASH, novel biomarkers or methods are urgently needed to improve the diagnosis of NASH. Meanwhile, the mechanisms underlying the development of NAFLD remain unknown. In this study, we systematically analyzed DEGs and DMGs in the progression of NAFLD. Functional enrichment analysis was conducted based on DEGs and DMGs. Several pathways were screened out, indicating the potential mechanisms involved in the development of NAFLD. Subsequently, based on DEGs, DMGs, and machine learning methods, six models that differentiate NASH from NAFL were constructed and compared. The LR model outperformed other models and was determined as the final diagnostic model. 

NAFLD is a multifactorial disease, and exploring the molecular mechanisms and the functional pathways based on the accumulating transcriptional and epigenetic alterations might facilitate the understanding of NAFLD development. In the aspect of DEGs, the functional enrichment results indicated that oxidoreductase-related activity and the PPAR signaling pathway might be involved in the whole progression of NAFLD, while extracellular-matrix-related activity might only participate in the latter period of NAFLD, the transition from NAFL to NASH. Oxidative stress is defined as an imbalance between the production of reactive oxygen species (ROS) and the scavenging capacity of the antioxidant system. Previous studies have suggested a central role of oxidative stress in the transition from NAFL to NASH mediated by increased ROS production, which could lead to the reprogramming of hepatic lipid metabolism, changes in insulin sensitivity, and modulation of inflammation by interacting with innate immune signaling [17]. PPARs are a group of nuclear regulatory factors that provide fine tuning for key elements of glucose and fat metabolism and regulate inflammatory cell activation and fibrotic processes, all of which determine NASH progression [18]. Currently, several PPAR agonists such as PPARα agonist Wy14643, and PPARβ/δ agonists GW501516, GW0742, and MBX-8025 have been reported to be used in the treatment of NAFLD in experimental and clinical studies and for developing dual and pan-PPAR agonists, and might have a broader and more efficacious therapeutic potential in the future [19]. In addition, the extracellular matrix is a multi-molecule complex structure composed of collagen, elastin fibers, and structural glycoproteins, and has proven to be closely related to progressive fibrosis and inflammation in NASH [20]. Of note, oxidative stress and inflammation can lead to the excessive production of extracellular matrix in liver diseases, which is consistent with our enriched results in the continuous periods of NAFLD [21]. 

For DMGs, the functional analysis revealed that the development of NAFLD was primarily related to tight junction, pathogenic *Escherichia coli* infection, and GTPase-related activity. Tight junctions are intercellular adhesion complexes in the epithelia and endothelia that control paracellular permeability, playing a vital role in architecture and homeostasis in the liver [22,23]. The disruption of tight junctions can impair intestinal permeability, and the subsequent increased gut microbial translocation can lead to the inflammatory pathway involved in NASH development [24]. Previous studies have demonstrated that NAFLD contains a disease-specific gut microbiome, and alteration of the gut microbiota plays a significant role in its progression to NASH and cirrhosis [25,26]. An increase in *Escherichia coli* was observed in patients with advanced NASH fibrosis, which was consistent with our enrichment results [27]. Furthermore, the translocation of intestinal *E. coli* NF73-1 into the liver was found to be responsible for the high hepatic M1 population in a mice model, which further aggravated liver injury, leading to disturbance of the hepatic triglyceride metabolism and, eventually, NAFLD progression [28]. GTPases are conserved regulators of cell motility, polarity, adhesion, cytoskeletal organization, proliferation, and apoptosis, but the role of GTPases in the NAFLD progression remains unclear [29].

NAFLD is an umbrella term that comprises a continuum of liver abnormalities, ranging from NAFL to NASH [30]. NASH is defined as a more serious stage of NAFLD and has higher risks of liver and non-liver related adverse outcomes compared with those patients with NAFL alone [31]. Thus, the prompt and accurate diagnosis of NASH is of extreme significance in clinical practice. Several biomarkers have been developed in the past decades, but their performances vary across studies. The plasma cytokeratin 18 (CK18) fragment level, a marker of hepatocyte apoptosis, has been extensively evaluated in steatohepatitis, while its limited sensitivity of 58% (51–65%) makes it inadequate as a screening test for staging NASH [32,33]. Serum metabolomics has identified pyroglutamate as a diagnostic biomarker for NASH, with a sensitivity and specificity of 72% and 85%, respectively [34]. The intrahepatic thrombospondin 2 (*THBS2*) expression level has shown an AUROC of 0.915 for diagnosing NASH [35]. Six differentially methylated CpG sites in peripheral blood leukocytes can be potentially used as diagnostic biomarkers for differentiating NASH from NAFL, with AUCs ranging from 0.689 to 0.882 [36]. However, most previous studies have been merely based on a single omics platform. In this study, we adopted six machine learning methods, including LASSO, LR, RF, SVM, XGBoost, and KNN, to build a diagnostic model based on inversely related methylation-transcription genes. The LR model, consisting of *ApoF*, *THOP1*, and *BICC1*, saw the highest AUC (0.8819, 95%CI: 0.8128–0.9511). Although the LR model exhibited a moderate specificity of 64.71%, it could distinguish NASH from NAFL with a sensitivity of 97.87%. Furthermore, the integration of multi-omics data can avoid the randomness of single omics data and improve the diagnostic capacity of disease phenotypes, and hence aid in patient stratification [37,38].

In addition to the stable diagnostic ability, the LR model generated based on methylation-transcription data also possessed the following traits. DNA methylation alterations are highly reversible and change in response to environmental and lifestyle experiences such as diet, obesity, and physical activities [12]. Likewise, studies have indicated that NASH may regress to NAFL after treatment. Weight loss has the strongest association with histologic improvement in NASH, and the methylation patterns of NASH are also altered after bariatric surgery [31,39]. As a result, the LR model could monitor the dynamic disease conditions in real time in terms of the similar reversibility, either during the disease transition or after treatments. Moreover, the LR model also possesses a potential early predictive capacity, as DNA methylation alterations are inheritable and can be transferred to the next generation. A previous study has shown that a high-fat and high-cholesterol Western diet (WD)-induced maternal hypercholesterolemia increases the male offspring risk for NAFLD and metabolic diseases, which is related to the decreased *ApoB* gene expression regulated by DNA hypermethylation [40]. 

Furthermore, the underlying mechanism for the progression of NAFLD is complex and multifactorial, and the model genes selected by the LR model might help in improving the understanding of disease pathogenesis. Thus, we explored the expression pattern of the three genes in the mouse model. Apolipoprotein F (*ApoF*) is a minor apolipoprotein mainly involved in cholesterol transportation by inhibiting cholesteryl ester transfer protein (CETP) activity with LDL [41,42]. The gene BicC family RNA-binding protein 1 (*BICC1*) encodes an RNA-binding protein that was firstly identified in Drosophila melanogaster and was later shown to have important roles in vertebrate development and embryogenesis [43]. Unfortunately, no meaningful results were obtained for the two genes in the mouse model. We speculated that the main reason was the species difference; *ApoF* and *BICC1* may not have equally important functions in the disease progression in mice. In this study, the gene *ApoF* was expressed at lower levels in NASH patients compared with NAFL patients. Similarly, Liu et al. found that hepatic *ApoF* mRNA levels were decreased by high fat, cholesterol-enriched diets, and *ApoF* was subject to negative regulation by agonist-activated LXR or PPARα nuclear receptors binding to a regulatory element ~1900 bases 5′ to the *ApoF* promoter [44]. The concentration of protein *ApoF* in serum has also been quantified to decrease across NAFLD stages in a previous study, which was consistent with our study [45]. In addition, recent studies have revealed that *BICC1* might be involved in the immune response in the tumor microenvironment by affecting immune cells, especially macrophages, and the overexpression of *BICC1* was closely related to the poor prognosis in tumors such as gastric cancer and oral cancer, as well as multiple functional pathways such as focal adhesion and ECM-receptor interaction, which were also enriched in our study [46,47]. Thus, the similar higher expression pattern of *BICC1* in NASH patients suggests its potential role in the immune response in NAFLD development.

Unlike *ApoF* and *BICC1*, a similar different expression pattern of *THOP1* across species was observed, suggesting its potential key role in the development of NAFLD. Thimet oligopeptidase (*THOP1*) is a metallopeptidase widely distributed in mammalian tissues, initially purified from the soluble fraction of rat brain homogenates in 1983 [48]. Apart from the role in major histocompatibility class I (MHC-I) antigen presentation, *THOP1* was recently reported to be involved in energy metabolism regulation [49]. Gewehr et al. found that the *THOP1* null (*THOP1*−/−) mice gained 75% less body weight and showed neither insulin resistance nor non-alcoholic fatty liver steatosis (NAFLS) when compared with wild-type (WT) mice after 24 weeks of being fed a hyperlipidic diet (HD), and also observed a higher adipose tissue adrenergic-stimulated lipolysis in *THOP1*−/− mice [50]. Furthermore, specific genes and microRNAs involved in obesity and adipogenesis were differentially modulated in the liver and adipose tissue of *THOP1*−/− mice. An increased expression level of PPAR-γ, which was also enriched in our study, was observed in *THOP1*−/− mice fed the HD when compared with either *THOP1*−/− mice fed a standard diet (SD) or WT mice fed the HD [50]. Altogether, previous studies have suggested that *THOP1* could be a therapeutic target for controlling obesity and associated diseases such as insulin resistance and NAFLD.

Potential limitations of the present study should be noted. The biological mechanisms of *THOP1* in the progression of NAFLD remain to be explored. In addition, the study was based on research data from the public database, which might induce selection bias. Thus, a multicenter and large-scale study should be conducted to further validate our findings.

## 5. Conclusions

In conclusion, we systematically explored the potential mechanisms and pathways in the progression of NAFLD by tracing the flowing information in both the transcriptome and epigenome, and a diagnostic model based on the combination of gene expression and methylation data was built to differentiate NASH from NAFL. To the best of our knowledge, this is the first diagnostic model that employed both transcriptional and epigenetic data that can provide a robust diagnostic ability. Further studies remain to be implemented to explore the pathogenesis of NAFLD and refine patient stratification to benefit NAFLD populations.

## Figures and Tables

**Figure 1 biomedicines-11-00970-f001:**
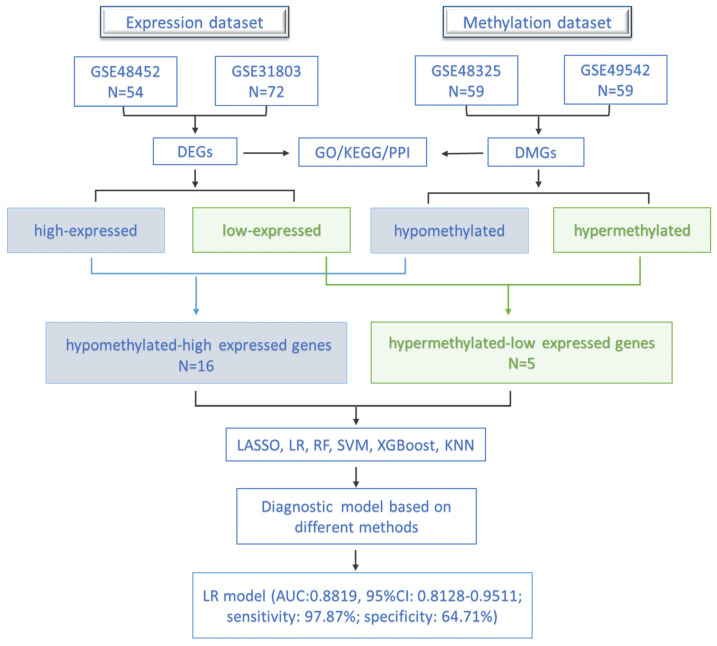
The workflow of this study.

**Figure 2 biomedicines-11-00970-f002:**
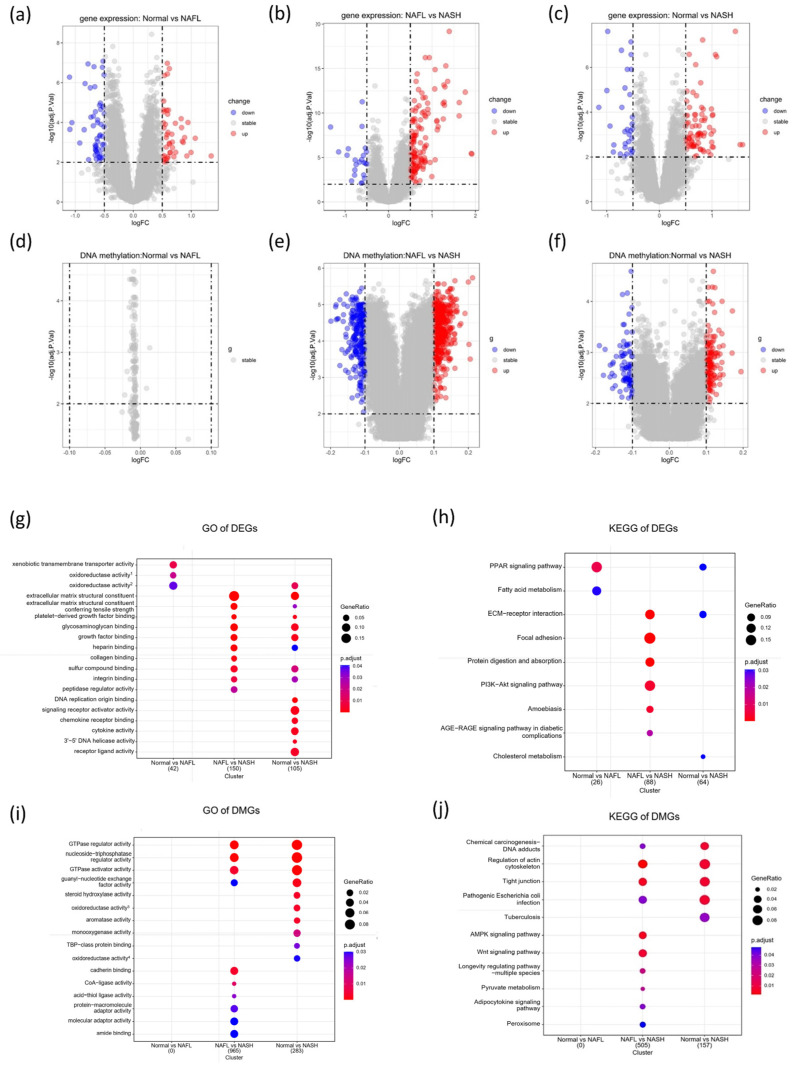
Identification of DEGs and DMGs in the progression of NAFLD. (**a**) Identification of DEGs in the normal vs. NAFL group; (**b**) identification of DEGs in the NAFL vs. NASH group; (**c**) identification of DEGs in the normal vs. NASH group; (**d**) identification of DMGs in the normal vs. NAFL group; (**e**) identification of DMGs in the NAFL vs. NASH group; (**f**) identification of DMGs in the normal vs. NASH group; (**g**) GO results of DEGs; (**h**) KEGG results of DEGs; (**i**) GO results of DMGs; (**j**) KEGG results of DMGs. Oxidoreductase activity^1^: oxidoreductase activity, acting on paired donors, with oxidation of a pair of donors, resulting in the reduction of molecular oxygen to two molecules of water; oxidoreductase activity^2^: oxidoreductase activity, acting on paired donors, with incorporation or reduction of molecular oxygen; oxidoreductase activity^3^: oxidoreductase activity, acting on paired donors, with incorporation or reduction of molecular oxygen, reduced flavin or flavoprotein as one donor, and incorporation of one atom of oxygen; oxidoreductase activity^4^: oxidoreductase activity, acting on paired donors, with incorporation or reduction of molecular oxygen, NAD(P)H as one donor, and the incorporation of one atom of oxygen.

**Figure 3 biomedicines-11-00970-f003:**
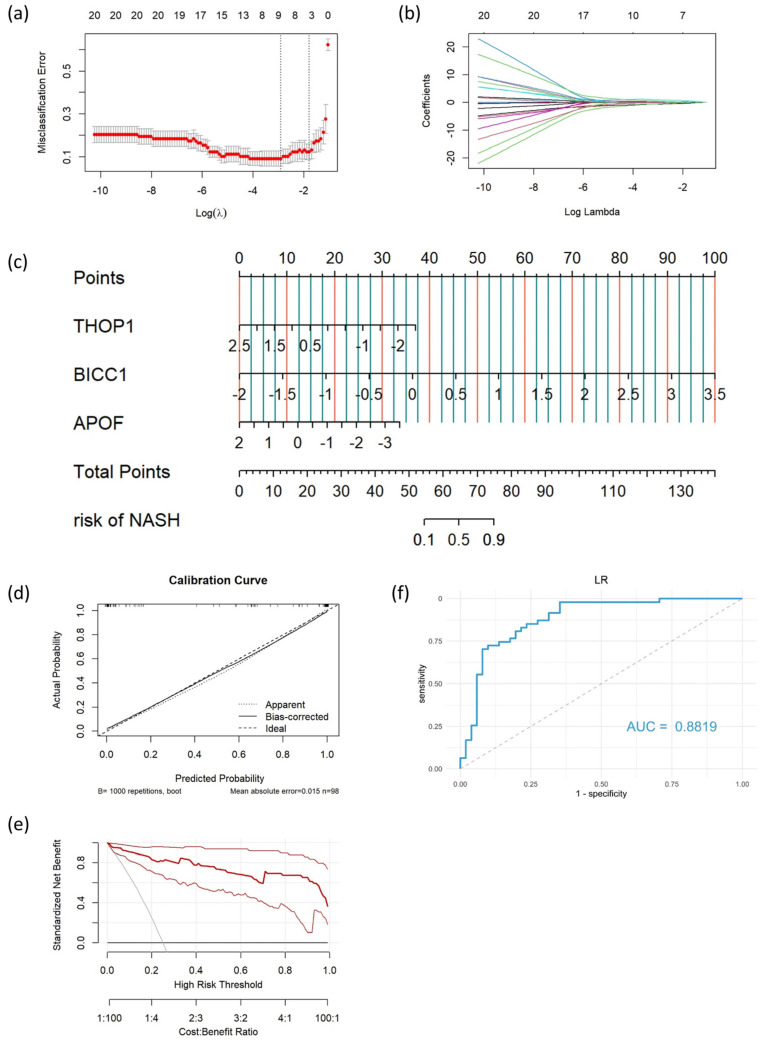
LASSO feature selection and the building process of the LR model. (**a**) The cross-validation plot for the optimal λ selection. The x-coordinate is the logarithm function of the penalty coefficient λ, and the y-coordinate is the misclassification error. (**b**) Plot for LASSO regression coefficients over different values of λ. As λ changes, the coefficients of the variables are compressed to zero. (**c**) Nomogram of the LR model. (**d**) Calibration curve of the LR model. The dashed line at 45° represents perfect prediction. (**e**) Decision curve analysis curve of the LR model. The bold red curve shows the benefit net of the LR model at different risk thresholds, while the curves on both sides represent its 95% confidence interval. (**f**) Receiver operating characteristic curve of the LR model.

**Figure 4 biomedicines-11-00970-f004:**
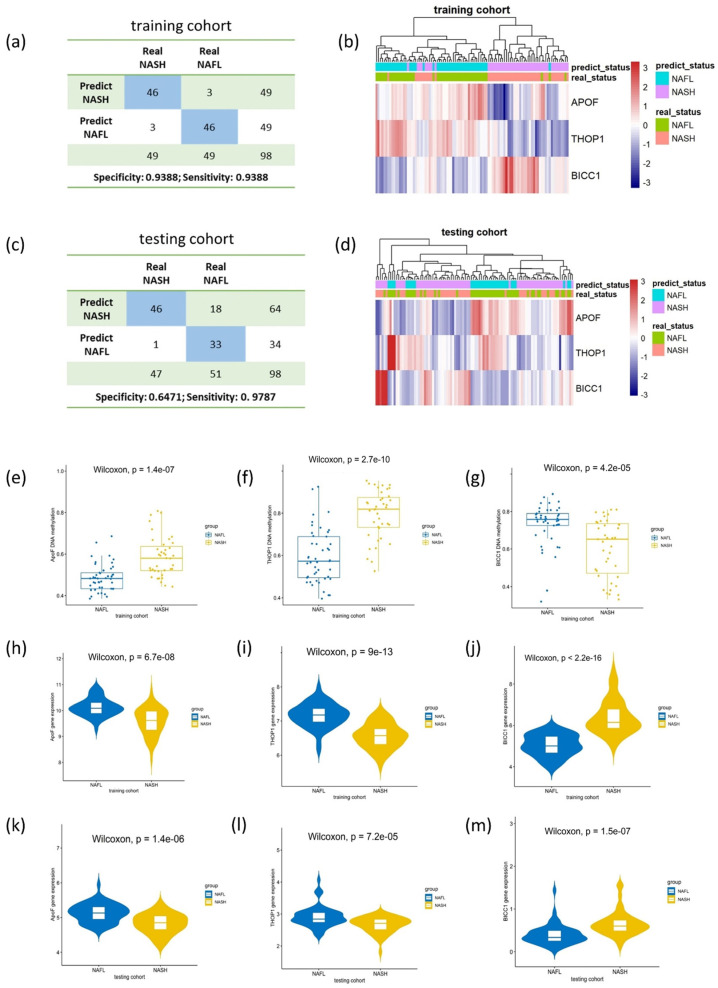
The performance of the LR model. (**a**,**c**) Confusion matrices of binary results of the LR model for training and testing cohorts. (**b**,**d**) Unsupervised hierarchical clustering of three model genes for the LR model in training and testing cohorts. (**e**–**g**) Scatter plots of the DNA methylation status of *ApoF*, *THOP1*, and *BICC1* in the training cohort. (**h**–**j**) Violin plots of the gene expression patterns of *ApoF*, *THOP1*, and *BICC1* in the training cohort. (**k**–**m**) Violin plots of the gene expression patterns of *ApoF*, *THOP1*, and *BICC1* in the testing cohort.

**Figure 5 biomedicines-11-00970-f005:**
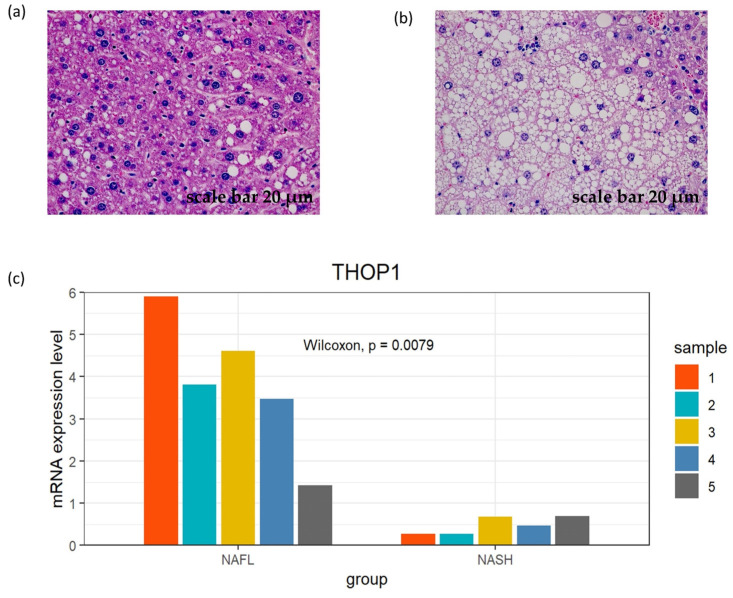
The expression pattern of *THOP1* in the mouse model. (**a**) Liver biopsy of NAFL sample 1 with hematoxylin-eosin staining, scale bar 20 μm. (**b**) Liver biopsy of NASH sample 1 with hematoxylin-eosin staining, scale bar 20 μm. (**c**) Bar plot of the gene expression pattern of *THOP1* in the mouse model.

**Table 1 biomedicines-11-00970-t001:** The detailed information of the included datasets.

	GEO ID	Platform	Mean Age (Years)	Sex (Male/Female)	Viral Hepatitis	Alcohol Use *	Sample Type	Sample Number		
								Normal	NAFL	NASH
Gene expression	GSE48452	GPL11532	46	11/43	0	0	liver biopsy	28	9	17
	GSE31803	GPL570	N/A	N/A	0	0	liver biopsy	0	40	32
DNA methylation	GSE48325	GPL13534	48	15/44	0	0	liver biopsy	34	10	15
	GSE49542	GPL13534	N/A	N/A	0	0	liver biopsy	0	35	24

*: >20 g/day for women, >30 g/day for men; N/A: Not applicable.

**Table 2 biomedicines-11-00970-t002:** The performance of the six models.

		Specificity	Sensitivity	PPV	NPV	AUC (95%CI)
LR	Training	93.88%	93.88%	93.88%	93.88%	0.9792 (0.9575–1.0000)
	Testing	64.71%	97.87%	71.88%	97.06%	0.8819 (0.8128–0.9511)
SVM	Training	93.88%	89.80%	93.62%	90.20%	0.9775 (0.9556–0.9994)
	Testing	66.67%	97.87%	73.02%	97.14%	0.8623 (0.7868–0.9378)
KNN	Training	93.88%	93.88%	93.88%	93.88%	0.9842 (0.9677–1.0000)
	Testing	70.59%	95.74%	75.00%	94.74%	0.8502 (0.7707–0.9298)
RF	Training	100.00%	100.00%	100.00%	100.00%	1.0000 (1.0000–1.0000)
	Testing	62.75%	93.62%	69.84%	91.43%	0.8454 (0.7695–0.9214)
XGBoost	Training	100.00%	100.00%	100.00%	100.00%	1.0000 (1.0000–1.0000)
	Testing	66.67%	89.36%	71.19%	87.18%	0.8256 (0.7455–0.9058)
LASSO	Training	95.92%	89.80%	95.65%	90.38%	0.9733 (0.9476–0.9991)
	Testing	72.55%	76.60%	72.00%	77.08%	0.8052 (0.7192–0.8911)

LR: logistic regression; LASSO: least absolute shrinkage and selection operator; KNN: K-nearest neighbor; RF: random forest; SVM: support vector machine; XGBoost: extreme gradient boosting; PPV: positive predictive value; NPV: negative predictive value; AUC: area under curve; CI: confidence interval.

## Data Availability

The data presented in this study are available upon request from the corresponding author.

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
