# Peer review of "Transcriptional and Epigenetic Alterations in the Progression of Non-Alcoholic Fatty Liver Disease and Biomarkers Helping to Diagnose Non-Alcoholic Steatohepatitis"

_biomedicines, 2023, doi:10.3390/biomedicines11030970_

Round 1

Reviewer 1 Report

„Transcriptional and epigenetic alterations in the progression of non-alcoholic fatty liver disease and biomarkers helping to diagnose non-alcoholic steatohepatitis”.

From the transcriptome and epigenome authors determined the diagnostic model which can differentiate NASH from NAFL. This analysis offers insights into NAFLD and NASH, but this approach is not without limitations. Biological samples in Gene Expression Omnibus have limitations in terms of description of samples. Moreover, some details that may present confounding variables are: the co-morbidities in patients, differing stages in fatty liver disease,  degree of hepatic fibrosis, medications, age, gender, and ethnicity. Furthermore, samples were taken under different conditions and the diagnoses were made by separate pathologists.

Although transcriptomic and epigenome studies can offer a global view of disease function and regulatory signaling using gene expression patterns, the causality necessitates more direct functional experimentation. Therfore, the authors investigated the expression pattern of the selected genes (ApoF, BICC1, and THOP1) in the mouse model, and suggested the potential key role of THOP1 in the development of NAFLD. In addition, the functional analysis revealed that the development of NAFLD was primarily related to the oxidoreductase-related activity, PPAR signaling pathway, tight junction and pathogenic Escherichia coli infection. This results offer a foundation to future studies in NAFLD and NASH. Thus, the manuscript is interesting for the Biomedicines readers.

Author Response

Response to Reviewer 1 Comments

Point 1: From the transcriptome and epigenome authors determined the diagnostic model which can differentiate NASH from NAFL. This analysis offers insights into NAFLD and NASH, but this approach is not without limitations. Biological samples in Gene Expression Omnibus have limitations in terms of description of samples. Moreover, some details that may present confounding variables are: the co-morbidities in patients, differing stages in fatty liver disease, degree of hepatic fibrosis, medications, age, gender, and ethnicity. Furthermore, samples were taken under different conditions and the diagnoses were made by separate pathologists.

Although transcriptomic and epigenome studies can offer a global view of disease function and regulatory signaling using gene expression patterns, the causality necessitates more direct functional experimentation. Therefore, the authors investigated the expression pattern of the selected genes (ApoF, BICC1, and THOP1) in the mouse model, and suggested the potential key role of THOP1 in the development of NAFLD. In addition, the functional analysis revealed that the development of NAFLD was primarily related to the oxidoreductase-related activity, PPAR signaling pathway, tight junction and pathogenic Escherichia coli infection. This results offer a foundation to future studies in NAFLD and NASH. Thus, the manuscript is interesting for the Biomedicines readers.

Response 1: Thank you very much for your time involved in reviewing the manuscript and your very encouraging comments on the merits. We also appreciate your clear and detailed feedback on the demerits of our work. We admit that the description of samples from Gene Expression Omnibus is limited. To be more rigorous, we have listed the basic information of datasets we adopted and annotated the not applicable items in Table 1. In addition, we have adopted strict criteria when filtering the datasets and we also excluded some samples in order to reduce the effects of confounding variables. The detailed description is as follows: Strict criteria were adopted when selecting the samples: patients without a clear histological diagnosis were excluded, and patients who received additional clinical treatments like bariatric surgery were also excluded.

Once again, thank you very much for your positive comments and valuable suggestions.

Reviewer 2 Report

Authors evaluated gene expression and methylation in previous database with NAFLD and NASH and constructed the prediction model for NASH with the combination of 3-genes expression. It was an interesting study. But several issues remained to be addressed.

1, In clinical, NASH is involved in NAFLD. In NASH, variety of disease progression might be found. Authors should clarify the staging of NASH in present study. Disease severity such as liver inflammation, liver fat degeneration or liver fibrosis should be considered in present study. 

2. In mice model, only THOP1 showed the significant difference. Authors should describe and discuss the reason why ApoF and BICC did not show significant difference in mice model. 

Author Response

Response to Reviewer 2 Comments

Reviewer #2: Authors evaluated gene expression and methylation in previous database with NAFLD and NASH and constructed the prediction model for NASH with the combination of 3-genes expression. It was an interesting study. But several issues remained to be addressed.

Point 1: In clinical, NASH is involved in NAFLD. In NASH, variety of disease progression might be found. Authors should clarify the staging of NASH in present study. Disease severity such as liver inflammation, liver fat degeneration or liver fibrosis should be considered in present study.

Response 1: Thank you for your professional guidance. We agree that the disease stage of NASH is of great importance. However, the specific histological information of each sample is not available because the data were taken from the public database. To make up for this deficiency, we have constructed a mouse model and clarified the NAFLD activity score (NAS) of each sample in the Table S1. The pictures taken from representative areas showing steatosis, lobular inflammation and hepatocyte ballooning were also provided (Figure S1).

Point 2: In mice model, only THOP1 showed the significant difference. Authors should describe and discuss the reason why ApoF and BICC1 did not show significant difference in mice model.

Response 2: Thank you very much for pointing this out and we have accordingly revised our manuscript. In the Results part, we have added the following descriptions: However, no meaningful findings were observed for the other two genes. In the Discussion part, we have added the following content: 

Apolipoprotein F (ApoF) is a minor apolipoprotein mainly involved in cholesterol transportation by inhibiting cholesteryl ester transfer protein (CETP) activity with LDL [41, 42]. The gene BicC family RNA-binding protein 1 (BICC1) encodes an RNA-binding protein that was firstly identified in Drosophila melanogaster and later shown to have important roles in vertebrate development and embryogenesis [43]. Unfortunately, no meaningful results were obtained for the two genes in the mouse model. We speculated that the main reason was the species difference, the ApoF and BICC1 may not have equally important functions in the disease progression in mice. In this study, the gene ApoF was expressed at lower levels in NASH patients compared to NAFL patients. Similarly, Liu et al. found that hepatic ApoF mRNA levels are decreased by high fat, cholesterol-enriched diets, and ApoF is subject to negative regulation by agonist-activated LXR or PPARα nuclear receptors binding to a regulatory element ~1900 bases 5' to the ApoF promoter [44]. The concentration of protein ApoF in serum has also been quantified to decrease across NAFLD stages in a previous study, which was consistent with our study [45]. In addition, recent studies have revealed that BICC1 might be involved in the immune response in the tumor microenvironment by affecting immune cells, especially macrophages, and the overexpression of BICC1 was closely related to the poor prognosis in tumors like gastric cancer and oral cancer, as well as multiple functional pathways like focal adhesion and ECM-receptor interaction, which were also enriched in our study [46, 47]. Thus, the similar higher expression pattern of BICC1 in NASH patients suggests its potential role in the immune response in NAFLD development.

Reviewer 3 Report

I read the paper “Transcriptional and epigenetic alterations in the progression of non-alcoholic fatty liver disease and biomarkers helping to diagnose non-alcoholic steatohepatitiby Zhu et al.

The paper is well written. It is appropriately divided into sections and subsections. Statistical analysis is well performed.

Comment:

1.     As the author have reported in the discussion HCC development, it should be introduced as follow the main pathophysiological mechanisms underlying as they will be further mentioned (at least some of them). “The pathophysiological mechanisms underlying the development of Non-alcoholic fatty liver disease (NAFLD) and its progression to Non-alcoholic steatohepatitis (NASH) and cirrhosis are various and involve pro-inflammatory agents, oxidative stress, apoptosis, adipokines, JNK-1 activation, increased IGF-1 activity, immunomodulation, and alteration of the gut microbiota. Moreover, these mechanisms are thought to play a significant role in the development of NAFLD-related hepatocellular carcinoma.” (doi: 10.3390/biomedicines11020468).

2.     Introduction: it should be underlined that up to now no pharmacological treatment has been validated by current guidelines, thus making this pandemic more troublesome to handle. It is clear that diet and physical exercise is crucial, though some studies have reported novel therapeutical approach which should be validated with the use of PPARγ, and or SGLT2i (doi: 10.3390/biomedicines11020322).

3.     Another limitation is represented by the small sample size.

Author Response

Response to Reviewer 3 Comments

Reviewer #3: I read the paper “Transcriptional and epigenetic alterations in the progression of non-alcoholic fatty liver disease and biomarkers helping to diagnose non-alcoholic steatohepatiti” by Zhu et al.

The paper is well written. It is appropriately divided into sections and subsections. Statistical analysis is well performed.

Response: Thank you very much for your recognition of our work. We will continue our efforts with your encouragement.

Point 1: As the author have reported in the discussion HCC development, it should be introduced as follow the main pathophysiological mechanisms underlying as they will be further mentioned (at least some of them). “The pathophysiological mechanisms underlying the development of Non-alcoholic fatty liver disease (NAFLD) and its progression to Non-alcoholic steatohepatitis (NASH) and cirrhosis are various and involve pro-inflammatory agents, oxidative stress, apoptosis, adipokines, JNK-1 activation, increased IGF-1 activity, immunomodulation, and alteration of the gut microbiota. Moreover, these mechanisms are thought to play a significant role in the development of NAFLD-related hepatocellular carcinoma.” (doi: 10.3390/biomedicines11020468).

Response 1: Thanks for your valuable suggestion. The pathophysiological mechanisms referred above are largely associated with the functional enrichment results of our work. Based on your advice, we have broadened and enriched our discussion accordingly: Previous studies have demonstrated that NAFLD contains a disease-specific gut microbiome and alteration of the gut microbiota plays a significant role in its progression to NASH and cirrhosis. An increase of Escherichia coli was observed in patients with advanced NASH fibrosis, which was consistent with our enrichment results.

Point 2: Introduction: it should be underlined that up to now no pharmacological treatment has been validated by current guidelines, thus making this pandemic more troublesome to handle. It is clear that diet and physical exercise is crucial, though some studies have reported novel therapeutical approach which should be validated with the use of PPARγ, and or SGLT2i (doi: 10.3390/biomedicines11020322).

Response 2: Thanks for your professional suggestion. We have added the suggested content to the Introduction part of the manuscript and we hope that it is now more comprehensive: Besides, there is currently no approved specific therapy for NAFLD, and lifestyle changes such as exercise and dietary modifications remain the mainstream treatment of NAFLD.

Point 3: Another limitation is represented by the small sample size.

Response 3: Thank you for pointing this out and we completely agree that sample size is an important consideration in this study. To reduce the effect of the limited sample size, we searched the whole Gene Expression Omnibus database to obtain qualified samples at the very beginning of the study. To be more rigorous, we have adopted an external dataset as testing cohort and also conducted animal experiments to verify the reliability of the results. Despite that, the limitation represented by the sample size is inevitable, just as we noted at the end of the manuscript: a multicenter and large-scale study should be conducted to further validate our findings.

Round 2

Reviewer 2 Report

Revised manuscript was well addressed to reviewers' comments and well written. This in an interesting study.